# Surface Functionalization of Cotton Fabric with Fluorescent Dendrimers, Spectral Characterization, Cytotoxicity, Antimicrobial and Antitumor Activity

**Ivo Grabchev [1,\*], Desislava Staneva [2], Evgenia Vasileva-Tonkova [3] and Radostina Alexandrova [4]**

[1]   Faculty of Medicine, Sofia University "St. Kliment Ohridski", 1407 Sofia, Bulgaria
[2]   Faculty of Chemical Technology, University of Chemical Technology and Metallurgy, 1756 Sofia, Bulgaria; grabcheva@mail.bg
[3]   The Stephan Angeloff Institute of Microbiology, Bulgarian Academy of Sciences, 1113 Sofia, Bulgaria; evaston@yahoo.com
[4]   Institute of Experimental Morphology, Pathology and Anthropology with Museum, Bulgarian Academy of Sciences, Acad. G. Bonchev Street, Bl. 25, 1113 Sofia, Bulgaria; rialexandrova@hotmail.com
\*   Correspondence: i.grabchev@chem.uni-sofia.bg; Tel.: +359-0-2816-1319

**Abstract:** Poly(propylenimine) dendrimers from first and third generations modified with 1,8-naphthalimide units and their Zn(II) complexes have been investigated by absorption and fluorescence spectroscopy. These dendrimers have been deposited on a cotton cloth by the extraction method, producing yellow-colored textile materials. They have been characterized by defining their color coordinates $L^*a^*b^*$, XYZ and xy. The antimicrobial activity of dendrimers has been investigated *in vitro* against model gram-positive and gram-negative bacteria and yeasts. Being deposited onto the surface of cotton fabric, the studied dendrimers reduced bacterial growth and prevented the formation of bacterial biofilm. Anticancer and cytotoxicity activities have also been performed against HeLa and Lep-3 human tumor cell lines as model systems.

**Keywords:** dendrimer; metallodendrimer; 1,8-naphthalimide; antimicrobial activity; anticancer activity

## 1. Introduction

Dendrimers are very attractive macromolecules that, in the last years, have been studied extensively from different scientific aspects—biological, biomedical and regarding environment protection, photovoltaic and light-emitting devices, etc. [1–4]. New, highly bioactive dendrimers can be obtained by modification with bioactive monomer agents. Dendrimers are also considered to be an effective tool for anticancer therapy or drug delivery systems [5,6]. A new branch of the dendrimer studies deals with metallodendrimers having unique biological and biomedical activities. The incorporation of metal ions into the dendrimer structure opens interesting prospects for dendrimer chemistry and facilitates their enhanced biological activity [7,8]. Therefore, lately, the application of such compounds in medical chemistry as a new class of metal—containing biomolecules—has expanded significantly. Because of the high concentration of surface functional groups in their molecules, they have been investigated intensively, especially for applications as antibacterial and antifungal agents.

Derivatives of 1,8-naphthalimide are an interesting class of heterocyclic systems with promising biomedical and pharmacological activities. Many of them exhibit a high antibacterial, antifungal, antiviral anti-inflammatory or anticancer activity [9–11] or sensor properties [12].

In our laboratory, we have started studies aimed at combining the properties of dendrimer molecules with those of 1,8-naphthalimide fluorophores. For this purpose, various dendrimer structures have been peripherally modified with 1,8-naphthalimide units [13]. The incorporation of a larger number of biologically active 1,8-naphthalimide units into one dendrimer molecule results in an increase in their activity. The bioactivity based on the combination of the effects of a dendrimer matrix, 1,8-naphthalimide units and a Cu(II) or Zn(II) metal complex produced by a single molecular structure have been explored [14–19].

In the present study, we report on the antimicrobial and cytotoxic/anticancer activity of poly(propyleneimine) dendrimers from first and third generations modified with 4-amino-1,8-naphthalimide and their Zn(II) complexes, containing four or sixteen units, respectively. The antimicrobial activity of the dendrimers deposited onto the surface of a 100% cotton fabric has also been examined and discussed with regard to the potential applications of the modified textiles as antibacterial materials.

## 2. Experimental Part

### 2.1. Materials

Dulbecco's modified Eagle's medium (D-MEM) and fetal bovine serum were obtained from Gibco-Invitrogen (Loughborough, UK). Thiazolyl blue tetrazolium bromide (MTT) was purchased from Sigma-Aldrich Chemie GmbH (Taufkirchen, Germany). Dimethylsulfoxide (DMSO) and trypsin were obtained from AppliChem (Darmstadt, Germany). The antibiotics (Penicillin and Streptomycin) were from Lonza (Verviers, Belgium). All the other chemicals of the highest purity commercially available were purchased from local agents and distributors. All sterile plastics and syringe filters were obtained from Orange Scientific (Braine-l'Alleud, Belgium). *N,N-Dimethylformamide* (DMF) (ACS *spectrophotometric grade* of ≥99.8%) was used. *Dimethyl sulfoxide* (DMSO) for molecular biology (CAS, Sigma–Aldrich) has been used for antibacterial screening

### 2.2. Methods

The synthesis and characterization of poly(propylenimine) dendrimers (PPA) from the first D4 and third D16 generations modified with 4-amino-1,8-naphthalimide polypropileamine and their Zn(II) complexes have been described recently [20]. The chemical structures of metallodendrimers (Zn(D4)(NO$_3$)$_2$ and Zn$_4$(D16)(NO$_3$)$_8$) and the respective dendrimer ligands D4 and D16 (without Zn(II) ions) are presented in Scheme 1. The ChemBioDraw Ultra 14.0 software was used to depict the chemical structure of dendrimers presented in Scheme 1.

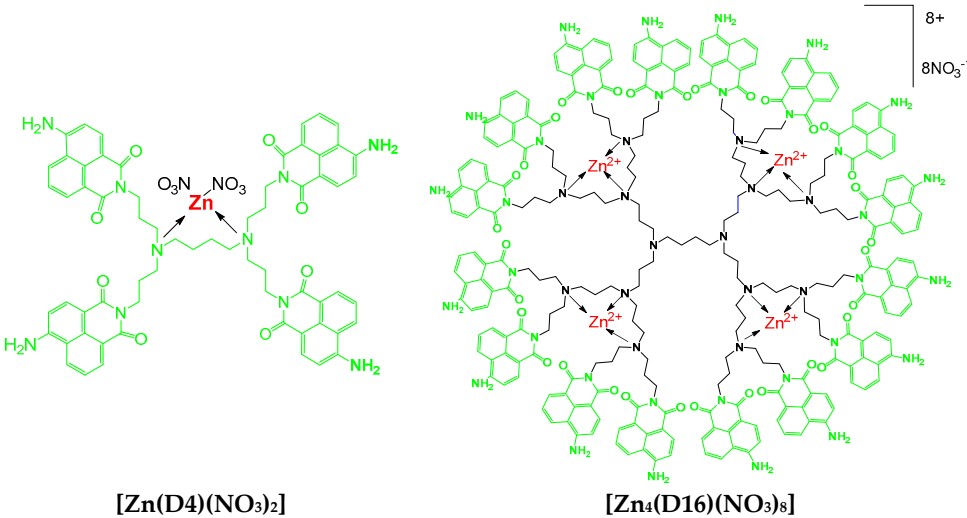

**[Zn(D4)(NO₃)₂]**          **[Zn₄(D16)(NO₃)₈]**

**Scheme 1.** The chemical structure of metallodendrimers.

### 2.2.1. Cotton Fabric Functionalization with Dendrimers D4 and D16 and Their Zn(II) Complexes

Of each dendrimer, 0.005 g was dissolved in 5 mL of a DMF–water 1:4 (v/v) solution. The cotton fabric sample (1 g) (weight: 140 g m$^{-2}$) was immersed in the solution at 25 °C for 30 min, washed with water and dried at ambient temperature.

### 2.2.2. Color Measurements

In order to determine how the presence of dendrimers D4 and D16 and their Zn(II) complexes affects the color of the cotton substrate, color measurements were carried out using a Spectraflash SF300 DATACOLOR apparatus (Datacolor International, New Jersey USA) and Micromatch 2000® software. The samples were measured under illuminant D65 using the 10° standard observer in the visible spectrum region 360–700 nm. The colour strength parameter (K/S) was used for the dyeing quality. In this case. the intensity of the color was expressed based on the Kubelka–Munk Equation (1) [21,22]:

$$K/S = (1 - R)^2/2R \tag{1}$$

where K is the light absorption coefficient of cotton fabric, S is the light scattering coefficient of cotton fabric and R is the reflectance of the cotton fabric expressed in fractional form.

The color difference ΔE*, which is a parameter to explain the difference between two colors, is estimated according to Equation (2) [23]:

$$\Delta E^* = [(\Delta L^*)^2 + (\Delta a^*) + (\Delta b^*)^2]^{1/2}, \tag{2}$$

where ΔL*, Δa* and Δb* are the difference in coordinates of two points present in the CIELab (L*, a* and b*) colour scale. In the case of the nonrated with dendrimers cotton fabric, ΔE* = 0.

### 2.2.3. Microbial Growth Inhibition Assay

Bacterial strains *Bacillus subtilis* ATCC 6633 and *Pseudomonas aeruginosa* 1390 and the yeasts *Candida lipolytica* 7618 were used as model cultures in our study (collection of the Institute of Microbiology, Sofia, Bulgaria). The growth inhibition of the strains by dendrimers D4 and D16 and their Zn(II) complexes (Zn(D4)(NO$_3$)$_2$ and Zn$_4$(D16)(NO$_3$)$_8$) was investigated in meat-peptone broth (MPB, pH 7.0) as a culture medium. Stock solutions of the investigated compounds in DMSO (0.5%) were prepared. From those, serial dilutions were made in test tubes with sterile MPB to final concentrations of 500, 350, 250, 125, 62.5, 31.25 and 15 μg/mL. The tubes were inoculated with 1% of each standardized inoculum of the test cultures and incubated at appropriate temperature for 24 h under shaking. The cell growth was assessed by measuring the turbidity at 600 nm (OD$_{600}$). The positive-control tubes contained MPB inoculated with test cultures, and the negative-control tubes contained MPB without microbial cells. The percent survival of the test cultures was determined on the basis of the positive control which was considered as 100%. The lowest concentration of the compounds that was able to inhibit visible growth of microbes was referred as Minimum Inhibitory Concentration (MIC). All assays were performed in triplicate.

### 2.2.4. Antimicrobial Test of Cotton Fabrics

The antimicrobial effect of the obtained modified cotton fabrics was tested against *B. cereus*, *P. aeruginosa* and *C. lipolytica* as model strains. The test tubes with sterile MPB were inoculated with each overnight grown microbial culture, and square-shape cotton specimens (10 mm × 10 mm) were inserted into the test tubes. The tubes with untreated cotton sample and the tubes without specimens were also prepared as controls. After 24 h of incubation at appropriate temperature, the specimens were removed, and the OD$_{600}$ was determined. To evaluate the antimicrobial activities of the samples, the reduction in microbial growth between the untreated and treated samples after incubation was compared.

### 2.2.5. Preparation of Cotton Fabrics for SEM

The adhesion and biofilm formation on cotton fabrics were assessed using scanning electron microscope (SEM). Sterile untreated and treated cotton fabrics were incubated 24 h in MPB inoculated with the cell suspension of *P. aeruginosa.* After incubation, the samples were washed with phosphate buffered saline, dried, coated with gold with Jeol JFC-1200 fine coater (Jeol Ltd., Tokyo, Japan) and investigated by SEM model Jeol JSM-5510 (Jeol Ltd., Japan).

### 2.2.6. Cell Cultures and Cultivation

The following cell lines were used as model systems in our study: HeLa (human cervical carcinoma) and Lep-3 (non-tumor human embryonic fibroblasts). The cell lines were obtained from the Cell Culture Collection of the Institute of Experimental Morphology, Pathology and Anthropology with Museum, Bulgarian Academy of Sciences.

The cells were grown as monolayer cultures in a DMEM medium supplemented with 5–10% fetal bovine serum, 100 U/mL penicillin and 100 µg/mL streptomycin. The cultures were kept in a humidified incubator (Thermo Scientific, HEPA Class 100) at 37 °C under 5% $CO_2$ in air. For routine passages, adherent cells were detached using a mixture of 0.05% trypsin and 0.02% Ethylenediaminetetraacetic acid (EDTA). The cell lines were passaged 2–3 times per week (1:2 to 1:3 split). The experiments were performed during the exponential phase of cell growth.

### 2.2.7. Cytotoxicity Assay

The cells were seeded in 96-well flat-bottomed microplates for cell culturing at a concentration of $1 \times 10^4$ cells/well. At the 24th hour, the culture medium was removed and changed with a media modified by different concentrations of the compound tested. Each concentration was applied in 6 to 8 wells. Samples of the cells grown in non-modified medium served as the control. The cytotoxicity of the complex against tumor (HeLa) and non-tumor (Lep-3) cell lines was evaluated using a colorimetric method based on the tetrazolium salt MTT (3-(4,5-dimethylthiazol-2-yl)-2,5-diphenyltetrazolium bromide) in accordance with Mossman's procedure. The cells were incubated for 3 h with the MTT solution (5 mg MTT in 10 mL DMEM) at 37 °C under 5% carbon dioxide and 95% air. The formed blue MTT formazan was extracted with a mixture of absolute ethanol and DMSO (1:1, vol/vol). The quantitative analysis (colorimetric evaluation of fixed cells) was performed by absorbance measurements in an automatic microplate reader at 540/620 nm (TECAN, SunriseTM, Austria).

### 2.2.8. Statistical Analysis

The data are presented as the mean of at least three replicates $\pm$ the standard error of the mean. Statistical differences between the control and treated groups were assessed using one-way analysis of variance (ANOVA) followed by Dunnett post hoc test. The results were considered significant when $p < 0.05$.

## 3. Results and Discussion

PPA dendrimers from first and third generations were modified whit 4-amino-1,8-naphthalimide units, thus obtaining new dendrimers with specific functional properties. In organic solvents, those dendrimers have a yellow color ($\lambda_A$ = 407–437 nm) and emit an intense yellow-green fluorescence with maxima in the spectral region of $\lambda_F$ = 507–541 nm. The basic photophysical characteristics of both dendrimers depend strongly on the solvents' polarity and exhibit a positive solvatochromism which can be explained by the fact that the excited 1,8-naphthalimide fluorophores molecules are better stabilized in polar organic solvents due to the stronger interactions with the dipoles of the solvents [20]. Figure 1a shows the absorption spectra of D4 in titration with Zn(II) ions as an example in DMF solution. As seen, the absorption maxima are shifted bathochromically from 432 nm (before the addition of Zn(II)) to 425 nm (after their addition). That displacement can be explained by a change in the polarization

of the chromophore system upon the complex formation. On the other hand, it can be seen that the absorption remains constant, which can be explained by the fact that Zn(II) ions do not form a complex directly with the 1,8-naphthalimide chromophore system. Figure 1b plots the fluorescence spectra of dendrimer D4 during the titration with Zn(II) ions. A slight decrease in the intensity of fluorescence emission has been recorded with an increase in Zn(II) concentration (about 10%), which has been accompanied by a bathochromic shift of the fluorescence maxima with 16 nm. As Figure 1b also shows, Zn(II) forms a complex with a dendrimer molecule at a 1:1 stoichiometry. In this case, the possible formation of the coordinate bond of Zn(II) is with the tertiary amino groups in the dendrimer. In the case of D16 as a ligand, it has been found that four Zn(II) ions form a complex with the tertiary amine groups of the core dendrimer molecule (Scheme 1).

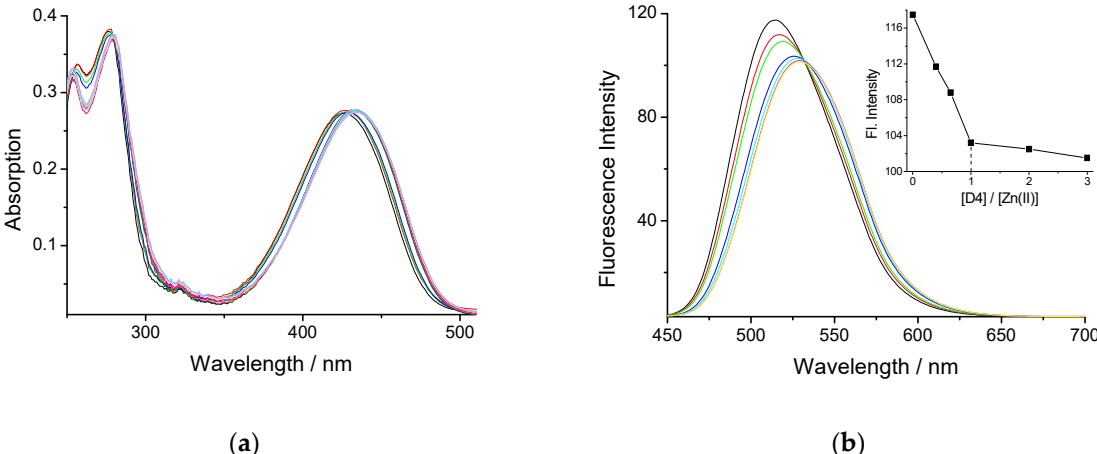

(**a**)  (**b**)

**Figure 1.** The absorption (**a**) and fluorescence (**b**) spectra of dendrimer D4 (c = 2 × 10$^{-5}$ mol/L in the presence of Zn(II) (c = 0 ÷ 6 × 10$^{-5}$ mol/L).

### 3.1. Color Characteristics of Cotton Fabric Modified with Dendrimers D4 and D16 and their Zn(II) Complexes

The textile samples dyed with dendrimer ligands D4 and D16 and their Zn(II) complexes have been analyzed by measuring their color characteristics by the CIE 1976 *L\*a\*b\** system, wherein each colour is represented by three coordinates *L\**, *a\** and *b\**. In this system *L\** represents lightness, *a\** represents redness if positive and greenness if negative and *b\** represents yellowness if positive and blueness if negative. The tristimulus values XYZ and chromaticity coordinates (x and y) have also been used to characterize the cotton fabric treated with different dendrimers. Table 1 summarizes the color characteristics of the textile materials investigated.

**Table 1.** The color characteristics of non-treated cotton fabric and cotton treated with dendrimers and metallodendrimers.

|  | L * | a * | b * | X | Y | Z | x | y |
|---|---|---|---|---|---|---|---|---|
| Cotton (control) | 92.83 | −0.06 | 2.67 | 78.26 | 82.52 | 84.90 | 0.3185 | 0.3361 |
| Cotton D4 | 86.98 | −1.99 | 41.81 | 65.44 | 69.96 | 33.55 | 0.3873 | 0.4141 |
| Cotton + Zn(D4)(NO$_3$)$_2$ | 85.53 | −0.58 | 37.85 | 63.32 | 65.05 | 34.65 | 0.3837 | 0.4063 |
| Cotton D16 | 89.16 | −2.00 | 31.63 | 69.77 | 74.58 | 45.04 | 0.3684 | 0.3938 |
| Cotton + Zn$_4$(D16)(NO$_3$)$_8$ | 87.04 | −1.06 | 34.89 | 65.98 | 70.09 | 39.09 | 0.3768 | 0.4003 |

The initial cotton fabric is white. After its treatment with dendrimers D4 and D16, the cotton fabrics acquire a dark green yellow color with different shades depending on the type and generation of the dendrimers and the number of zinc ions coordinates to dendrimer ligand. The textile treated with D6 has higher L* values and lower b* values against D4, and this relates to their complexes. In these cases, the tristimulus values XYZ increase and the chromaticity coordinates (x and y) decrease, which

give the perception of a brighter color. The bathochromic shift of the absorption and the fluorescence maxima upon the complex formation in solution influence the fabric color and are quantitatively characterized with the changes of coordinates. The greener shade and the lightness (L*, a* and X, Y) decreases after complex formation.

The estimation of the color difference ($\Delta E^*$) between the treated samples and untreated control sample has been determined by calculating the $\Delta E$ values (shown in Figure 2A). As seen from Figure 2, the cotton fabrics treated with D4 ($\Delta E^* = 39.9$) and D16 dendrimer ($\Delta E^* = 29.5$) have an intense color, as a better effect is being obtained in the case of D4 and its Zn(II) complex. This can be explained by the lower molecular weight of the dendrimers D4 and its complex and, hence, the better dye exhaustion from the bath and the better dye fixation on the surface of the cotton fabric. This is also confirmed by the K/S values shown in Figure 2B.

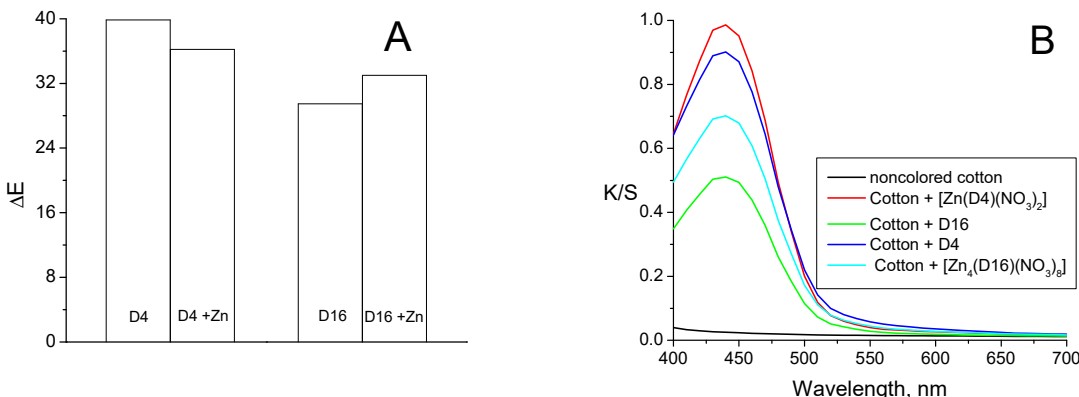

**Figure 2.** The color difference $\Delta E$ (**A**) and K/S (**B**) of cotton fabric treated with D4 and D16 and their Zn(II) complexes.

From Figure 2B, it can be seen that the noncolored cotton fabric has very low K/S values due to the low absorbance. After processing with the dendrimers, the K/S values increase. The results show that the cotton fabrics treated with metallodendrimers have a slightly more saturated color compared to D4 and D16. This fact reveals that the difference in the K/S values can be used as a criterion to detect zinc ions in aqueous environments. Thus, the textile materials treated with special fluorophores can be used for the development of heterogeneous sensors with the capacity to detect metal ions and protons in aqueous medium.

*3.2. Growth Inhibitory Activity*

The antimicrobial activity of the dendrimers and their Zn(II) complexes $Zn(D4)(NO_3)_2$ and $Zn_4(D16)(NO_3)_8$ was investigated quantitatively against *B. subtilis* (gram-positive) and *P. aeruginosa* (gram-negative) bacteria, and the antifungal activity was investigated against the yeast *C. lipolytica*. The results showed a low to moderate antimicrobial activity of the D4 and D16 samples depending on the strain. The D16 samples were found slightly more active that D4 samples, and $Zn(D4)(NO_3)_2$ and $Zn_4(D16)(NO_3)_8$ exhibited a slightly higher antimicrobial efficiency in comparison with the free ligands (Figure 3). For example, the D4 and D16 compounds reduced more than 80% the growth of the yeasts *C. lipolytica* at 250 µg/mL, while $Zn(D4)(NO_3)_2$ and $Zn_4(D16)(NO_3)_8$ completely inhibited the growth of the yeasts at this concentration. The complete inhibition of the growth of all test strains was observed at 500 µg/mL of the dendrimers and their Zn(II) complexes. According to criteria proposed by Holetz at al., MIC values $\leq 100$ µg/mL correspond with a good antimicrobial activity, 100 to 500 µg/mL with moderate and $\geq 500$ µg/mL with slight action [24]. Based on these criteria, the studied dendrimers exhibited moderately effective MIC values.

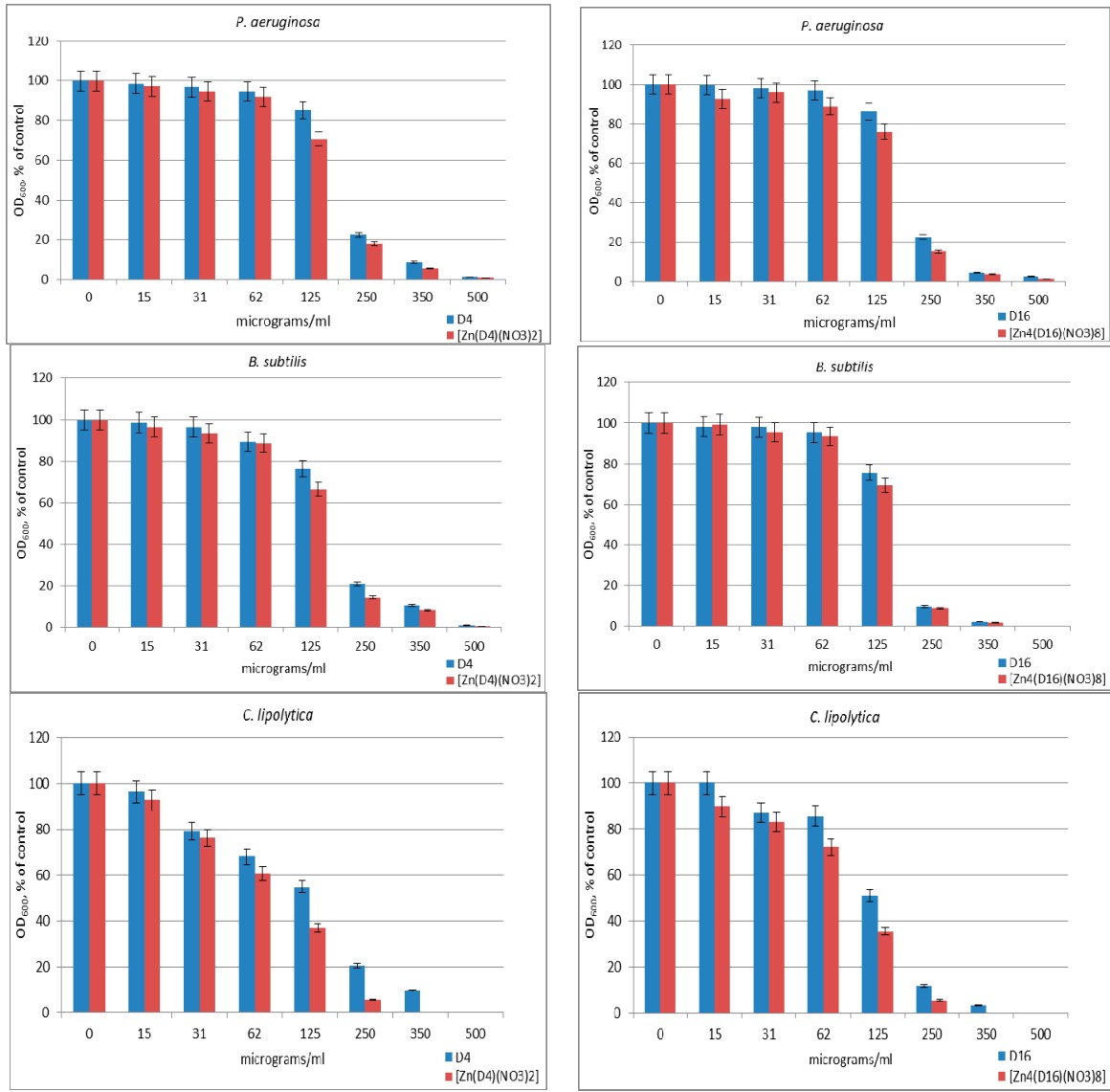

**Figure 3.** The effect of different concentrations of the dendrimer ligands (D4 and D16) and their complexes Zn(D4)(NO$_3$)$_2$ and Zn$_4$(D16)(NO$_3$)$_8$ on the growth of the test model strains.

The increased antimicrobial activity of the Zn(II) complexes could be explained by the Overtone's concept and chelation theory [25,26]. According to this theory, the lipid cell membrane favours the penetration of only particles soluble in lipids, which makes the lipid solubility a very important factor controlling the antimicrobial activity of the compounds. Chelation reduces the polarity of the Zn(II) and increases also the delocalization of π-electrons over the whole chelate complex, thus increasing the lipophilicity of the metallodendrimers. These changes are better expressed in Zn$_4$(D16)(NO$_3$)$_8$, probably leading to a higher solubility in the lipids compared to the Zn(D4)(NO$_3$)$_2$ complex. In this case, the higher lipophilicity can be explained with the higher number of Zn(II) ions in the metallodendrimers.

### 3.3. Antimicrobial Activity of Modified Cotton Fabrics

As hydrophobic and natural products, the cotton materials are widely used in households, industry and medicine, where they can be used for making hospital linen, wound dressings and other products. The structure of cotton fabrics favors the development of various bacteria, which requires the search for suitable substances to modify the cotton surface and to obtain materials with

antibacterial properties in order to prevent the formation of bacterial biofilms. The production of biofilms by bacteria can cause a resistance to various antibacterial agents. Thus, the inhibition of biofilm activity may be important for preventing infections and various other disorders. Overall, it has been shown that dendrimers are highly effective in biofilm inhibition and reduction [27].

The antimicrobial effect of cotton fabrics treated with the dendrimers and metallodendrimers was evaluated by a reduction of the growth of *B. cereus*, *P. aeruginosa* and *C. lipolytica* in meat-peptone broth. We found that cotton fabrics treated with the controls D4 and D16 did not reduce the growth of the test cultures, and a slight growth inhibition was determined by the complexes $Zn(D4)(NO_3)_2$ and $Zn_4(D16)(NO_3)_8$, which was better expressed in the yeast strain C. *lipolytica* (Figure 4). Several factors may considerably affect the antimicrobial effect of textiles such as the mechanical retention of microbial cells on textiles depending on the surface morphology; a dispersion of the antimicrobial material on the textile surface; and a variation of the hydrophobic/hydrophilic nature of textiles, which may influence the contact degree of the microbial inoculums with the textiles [28].

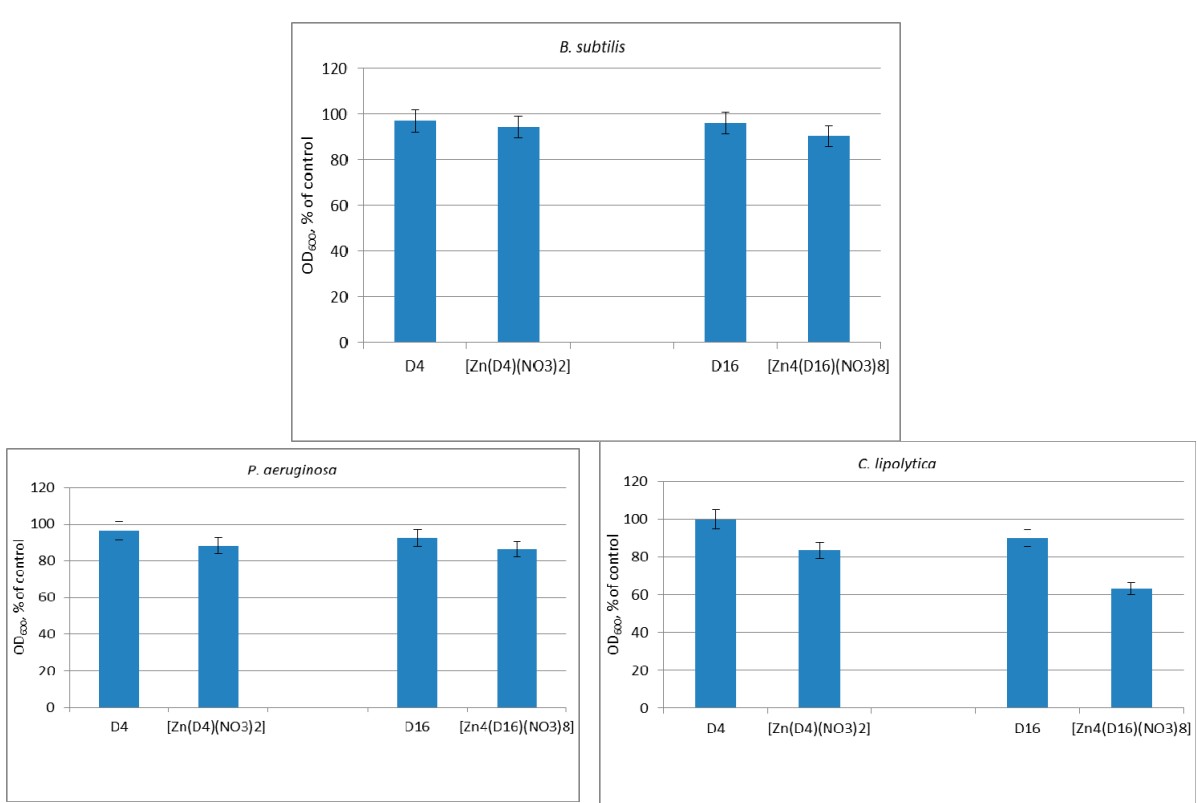

**Figure 4.** The effect of untreated cotton fabric D4 and D16 (controls) and cotton fabrics treated with $Zn(D4)(NO_3)_2$ and $Zn_4(D16)(NO_3)_8$ on the growth of the test microbial strains.

Dendrimers are water-insoluble, and they are firmly attached to the cotton surface mainly by hydrogen bonds and Van der Waals interactions. They cannot release from the cotton surface, and therefore, their contact with the bacteria in the solution is small, which indicates a higher viability of the bacteria in solution, but they prevent deposition and biofilm formation of bacteria on the cotton surface. This effect is well-illustrated in the SEM study of the surface of cotton fabrics treated with bacteria.

### 3.4. SEM Observations

The impact of dendrimers on the prevention of bacterial film formation has been investigated after their deposition onto 100% cotton fabric at 0.5% dendrimer concentration. Figure 5 shows a SEM micrograph of a cotton fabric treated with $Zn_4(D16)(NO_3)_8$. The uniform distribution of the

metallodendimeter over the cotton surface is clearly visible. As seen, the metallodendrimer molecules have almost a spherical shape and are located over the surface of the textile material.

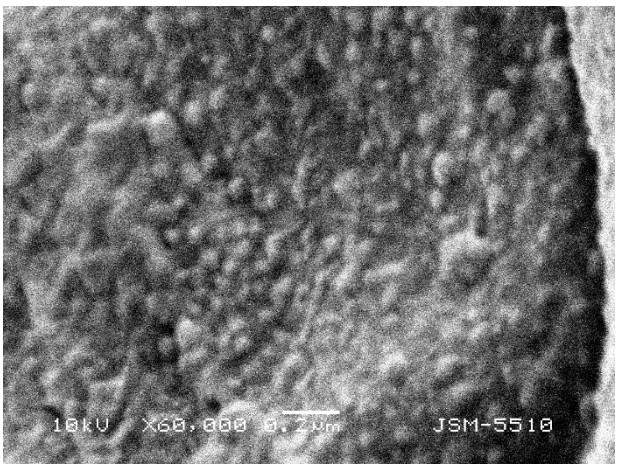

**Figure 5.** A SEM image of a cotton fabric treated with $Zn_4(D16)(NO_3)_8$.

The micrographs of a cotton fabric treated with dendrimer ligand D16 (Figure 6A) and a cotton fabric treated with its $Zn_4(D16)(NO_3)_8$ (Figure 6B) are presented in Figure 6.

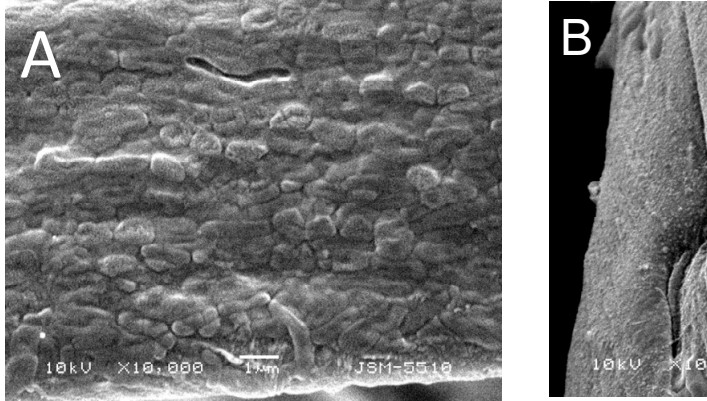
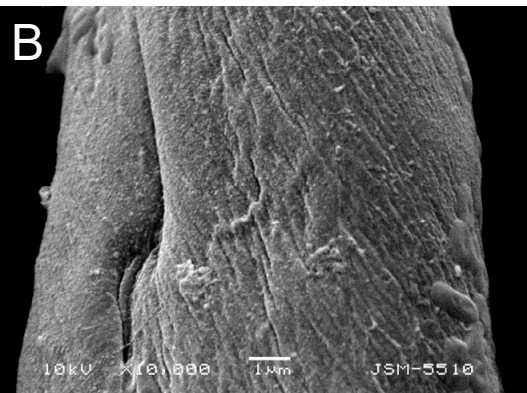

**Figure 6.** SEM images of cotton fabrics tested against *P. aeruginosa* at magnification ×10000: (**A**) the biofilm on cotton textile treated with D16 and (**B**) the absence of biofilm on a cotton textile treated with $Zn_4(D16)(NO_3)_8$.

The results indicate that a well-expressed, stable biofilm with *P. aeruginosa* colonies is formed on the surface of a cotton fabric treated with D16 (Figure 6A). The micrograph presented in Figure 6B shows the uniform distribution of the dendrimer on the surface of the cotton fabric thread with $Zn_4(D16)(NO_3)_8$. This thin film prevents biofilm formation and the clustering of bacteria; only single bacteria attached onto the cotton surface are observed. These results indicate that, despite the low activity of $Zn_4(D16)(NO_3)_8$ against the tested bacterial strains in solution after its deposition on cotton textile, it exhibits its antibacterial activity by protecting the treated cotton fabric from the available bacteria in the solution.

### 3.5. In Vitro MTT Cytotoxicity Assay

The ability of $Zn_4(D16)(NO_3)_8$ to influence the viability and growth of HeLa (human cervical carcinoma) and Lep-3 (non-tumor human embryonic fibroblasts) cells has been tested. The cytotoxic/anticancer activity of the compound has been evaluated using a 3-(4,5-dimethylthiazol-2-yl)-2,5-diphenyltetrazolium bromide (MTT) test. The results have been

analysed by means of cells viability after their treatment for 24 h and 48 h with $Zn_4(D16)(NO_3)_8$ at different concentrations (1.71, 4.28, 8.55, 17.12, 34.23 and 68.46 μM/mL).

The results obtained revealed that the examined compound decreased in a time- and concentration-dependent manner viability and/or proliferation of the treated cells (Figure 7). Non-tumor Lep-3 cells exhibited a lower sensitivity to the cytotoxic activity of $Zn_4(D16)(NO_3)_8$ than HeLa cancer cells when the compound was applied at a concentration range of 1.71 to 34.23 μM/mL. Administered at a concentration of 68.46 μM, the complex was more toxic for Lep-3 cells as compared to HeLa cells (Figure 8).

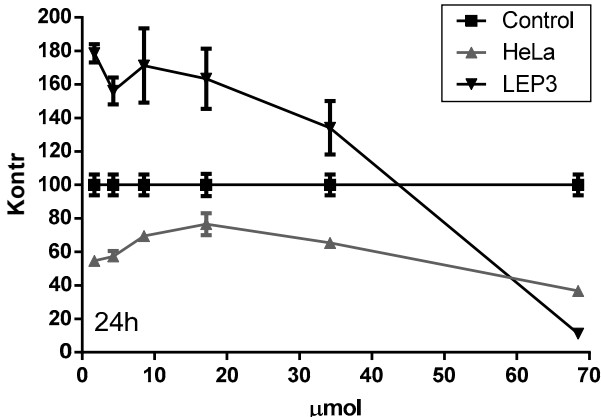

**Figure 7.** The effect of $Zn_4(D16)(NO_3)_8$] on the viability and/or proliferation of tumor (HeLa) and non-tumor (Lep-3) cells after a 24-h treatment period.

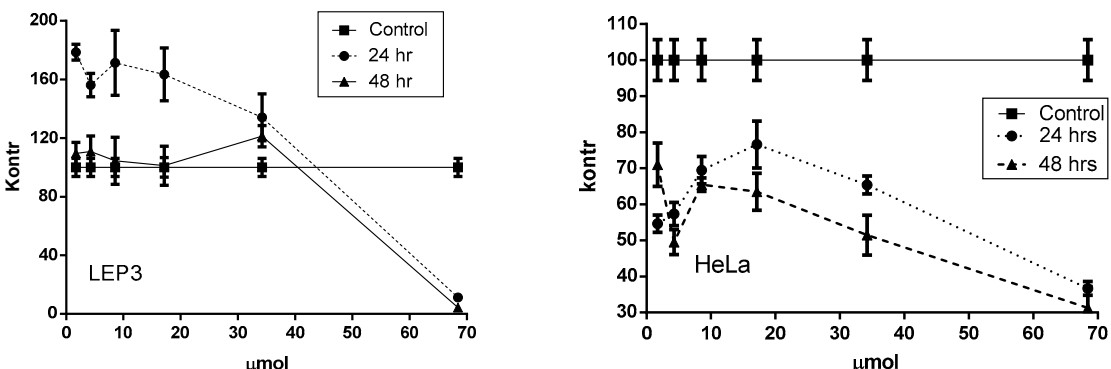

**Figure 8.** The viability of tumor (HeLa) and non-tumor (Lep-3) human cells cultured for 24 h and 48 h in the presence of $Zn_4(D16)(NO_3)_8$.

## 4. Conclusions

The influence of the concentration of Zn(II) ions on the spectral properties of two poly(propylenimine) dendrimers from first (D4) and third (D16) generations modified with 4-amino-1,8-naftalimide units has been investigated in a DMF solution by absorption and fluorescence spectroscopy. The results have shown that Zn(II) ions coordinate in the dendrimer core forming Zn(II) complexes. Dendrimer ligands and their isolated stable Zn(II) complexes have been deposited on the cotton fabric to obtain textile materials with a uniform yellow color and fluorescence. The cotton fabrics have been investigated and their color characteristics (*L\*a\*b\**, XYZ and xy) were determined. The K/S values showed that the cotton fabrics treated with Zn(II) complexes had a slightly more saturated color compared to the dendrimer ligands D4 and D16. The antimicrobial activity of dendrimers and after their deposition on the cotton fabrics were investigated in solution. At dendrimer concentrations of 250–350 μg/mL, more than 90% inhibition of the growth the tested microorganisms was achieved. Also, the results have shown that the cotton fabric treated with metallodendrimers prevents bacterial

biofilm formation. The obtained antibacterial cotton fabrics could be used as textile materials for the preparation of wound dressings or medical textiles for use in medical clinical practices. The preliminary results have shown a promising cytotoxic/anticancer activity of $Zn_4(D16)(NO_3)_8$ for human HeLa carcinoma cells.

**Author Contributions:** Conceptualization, I.G.; methodology, D.S., E.V.T. and R.A.; textile materials treatment, D.S.; writing—original draft preparation, I.G and E.V.T.; writing—review and editing, D.S.; visualization, D.S.; supervision, I.G.; funding acquisition, I.G.

**Funding:** The authors acknowledge Grant no. KOCT, 01/3-2017, fund "Scientific Research", Ministry of Education and Science of Bulgaria.

**Conflicts of Interest:** The authors declare no conflict of interest.

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
