# Peer review of "Surface Functionalization of Cotton Fabric with Fluorescent Dendrimers, Spectral Characterization, Cytotoxicity, Antimicrobial and Antitumor Activity"

_chemosensors, doi:10.3390/chemosensors7020017_

Round 1
Reviewer 1 Report
The synthesis of D4 and D16 was done as reference 20. I would like to see NMR spectra and ESI mass of just synthesised and the reported one with key representative proton signal comparison as a structural confirmation as a small figure. It is okay to reproduce the NMR of reported one for comparison in main text and a short discussion on NMR and mass spectra.
Figure 3: I do understand that at 250 mg/mL of D16 and its zinc complex, growth inhibition observed. What happen beyond 250 mg/mL....per say 350 mg/mL, 500 mg/mL? Is there same effect or no effect? There need an investigation.
From figure 1 to Figure 8 (except figure 4,5), None of these reading have studied with multiple replicates, which are essential for cell viability, cytotoxic MTT assay, growth inhibition assays. This is missing with all the bar graphs which is without error bars.Â
Line 161-163 says about statistical analysis with standard deviation bar, however, I didn't find those.
Eventually, this made very hard for me to accept this antimicrobial and anti bacterial data.
Kindly repeat experiments and resubmit as new paper or as a revised form.
Author Response
The synthesis of D4 and D16 was done as reference 20. I would like to see NMR spectra and ESI mass of just synthesised and the reported one with key representative proton signal comparison as a structural confirmation as a small figure. It is okay to reproduce the NMR of reported one for comparison in main text and a short discussion on NMR and mass spectra.
The synthesis of dendrimers and their zinc complexes is described in ref. 20. The text has been modified for better clarity. In this paper we use these compounds and explore their novel properties such as bioactivity and antibacterial textile.
Figure 3: I do understand that at 250 mg/mL of D16 and its zinc complex, growth inhibition observed. What happen beyond 250 mg/mL....per say 350 mg/mL, 500 mg/mL? Is there same effect or no effect? There need an investigation.
Answer:
Growth of the strains was investigated in presence of 350 µg/mL and 500 µg/mL of the compounds and the results are indicated in the revised Figure 3.
Â
From figure 1 to Figure 8 (except figure 4,5), None of these reading have studied with multiple replicates, which are essential for cell viability, cytotoxic MTT assay, growth inhibition assays. This is missing with all the bar graphs which is without error bars.
Answer:
Error bars are included on all graphs in the revised manuscript.
Â
Line 161-163 says about statistical analysis with standard deviation bar, however, I didn't find those.
Eventually, this made very hard for me to accept this antimicrobial and antibacterial data.
Kindly repeat experiments and resubmit as new paper or as a revised form.
Â
Done
Reviewer 2 Report
 Â
Dear Editor, dear Authors,
The manuscript entitled: "Antimicrobial and anticancer activity of fluorescent 2 Zn(II) complexes of poly(propyleneamine) dendrimer 3 modified with 1,8-naphthalimides" addresses the in vitro biolgic activity of novel synthesized Zn dendrimers.
Â
It is a well-written manuscript, the findings are relevant, however the paper does not highlighted the connections of the study with the major topic of the Journal: chemosensing, chemo-detection, and so on.
Even the title should be reformulated, since it is not just a study on synthesis-characterization-in vitro activity, but gives good results about the use of a dendrimer-coated cotton surface to attract and destroy the microorganisms and the tumor cells. The photophysical properties of the dendrimers should be hghlighted in the abstract
Please highlight any other aspects approaching the "chemosensor" character of compounds or methods.
Â
Therefore I recommend a major revision; other, minor comments are as follows:
Â
Abstract: page 1 row 21 "dendrimers from first and third generations" some details should be given about the meaning of first and third generation dendrimers, for the readers who are not experts in dendrimer structures (the same comment for introduction, row 54)
Â
Page 2, row 48: Please reformulathe the phrase: "In our laboratory, we started studies targeted at combining the properties of dendrimer 48 molecules with those of 1,8-naphthalimide fluorophores...."
Â
Page Methods, row 74: the meaning of the abbreviation PPA is missing from this sentence.
Page 3, Scheme 1- please insert the name of the software which depicted the structures.
No data was given about the synthesis of Zn(D4)(NO3)2 and Zn4(D16)(NO3)8
Row 82 -the abbreviation DMF was not explained, and they are no information about the provider of the cotton fabric (if relevant).
Â
Row 89-data should be given about the manufacturer of Spectraflash SF300 90 DATACOLOR apparatus and Micromatch 2000® software (manufacturer, city, country)
In the methods section they are materials, reagents having poor data about manufacturer.
Â
97- how the reflectance of the cotton fabric was measured?
Â
Row 101-provenance of bacterial strains is missing
Â
Page 4, row 104- no explanation for the abbreviation DMSO; the source of the reagents in this section was not specified
Â
row 130- please provide details about Jeol JFC-1200 and Jeol JSM-5510 manufacturer.
Â
row 136- non-tumor (not nin-tumor), or normal cells- Attention to page 11, row 311 where Lep-3 is presented as tumor cell line
Â
row 137 - to which institution belongs IEMPAM – BAS, or of they are independent, please provide the complete name without abbreviation, city, country.
Â
row 138- the source of cell culture supplements should be inserted
Â
Page 5, row 180: please reformulate: "This observation is confirmed by examining the fluorescence dependence."..."
Â
Page 7, row 209- explain K/S, ΔE, and so on
Page 6-7 for the Chemosensors readers, it could be interesting to find out how the color characteristics of the studied materials can serve for the development of new chemosensors
Â
Page 11, row 312- MTT is a viability test, or the evaluation of growth inhibition, not an "anticancer "test
Â
Page 11, figure 7- they are not two cancer cell lines! What represents the black "control"line?
Â
Page 12, figure 8- which graph represents HeLa and which is for Lep-3. They are not "different human cancer cells..."
Â
Page 12, conclusions: the part of manuscript which link the manuscript with the journal's topic is not discussed here. Â
Â
Page 14, row 471: the reference no. 28 is unfinished
Author Response
Abstract: page 1 row 21 "dendrimers from first and third generations" some details should be given about the meaning of first and third generation dendrimers, for the readers who are not experts in dendrimer structures (the same comment for introduction, row 54)
Done
 Page 2, row 48: Please reformulathe the phrase: "In our laboratory, we started studies targeted at combining the properties of dendrimer  molecules with those of 1,8-naphthalimide fluorophores...."
Done
Page Methods, row 74: the meaning of the abbreviation PPA is missing from this sentence.
Done
Page 3, Scheme 1- please insert the name of the software which depicted the structures.
ChemBioDraw Ultra 14.0 software has been used to depict chemical structure of dendriemers presented in Scheme 1.
No data was given about the synthesis of Zn(D4)(NO3)2Â and Zn4(D16)(NO3)8
The synthesis of dendrimers and their zinc complexes is described in ref. 20. The text has been modified for better clarity. In this paper we use these compounds and explore their novel properties such as bioactivity and antibacterial textile.
Row 82 -the abbreviation DMF was not explained, and they are no information about the provider of the cotton fabric (if relevant).
Done in materials
 Row 89-data should be given about the manufacturer of Spectraflash SF300 90 DATACOLOR apparatus and Micromatch 2000® software (manufacturer, city, country)
Done
In the methods section they are materials, reagents having poor data about manufacturer.
97- how the reflectance of the cotton fabric was measured?
Тhe cotton cloth is placed in a special place at the DATACOLOR apparatus and irradiated with light.
Row 101-provenance of bacterial strains is missing
Was added
 Page 4, row 104- no explanation for the abbreviation DMSO; the source of the reagents in this section was not specified
Done in materials
 row 130- please provide details about Jeol JFC-1200 and Jeol JSM-5510 manufacturer.
Done
row 136- non-tumor (not nin-tumor), or normal cells- Attention to page 11, row 311 where Lep-3 is presented as tumor cell line
 row 137 - to which institution belongs IEMPAM – BAS, or of they are independent, please provide the complete name without abbreviation, city, country.
Done
 row 138- the source of cell culture supplements should be inserted
Done
 Page 5, row 180: please reformulate: "This observation is confirmed by examining the fluorescence dependence."..."
Done
 Page 7, row 209- explain K/S, ΔE, and so on
Done
Page 6-7 for the Chemosensors readers, it could be interesting to find out how the color characteristics of the studied materials can serve for the development of new chemosensors
Done
 Page 11, row 312- MTT is a viability test, or the evaluation of growth inhibition, not an "anticancer "test
Done
 Page 11, figure 7- they are not two cancer cell lines! What represents the black "control"line?
 Page 12, figure 8- which graph represents HeLa and which is for Lep-3. They are not "different human cancer cells..."
 Corrected
Page 12, conclusions: the part of manuscript which link the manuscript with the journal's topic is not discussed here. Â
The conclusion has been rewrittenÂ
Page 14, row 471: the reference no. 28 is unfinished
Done
Â
Round 2
Reviewer 1 Report
Kindly arrange Figure 7 and figure 8 in the proper sequence.
The manuscript is accepted for a publication.
Reviewer 2 Report
The manuscript was revised by the authors. I consider that the changes in the manuscript were satisfactory, therefore I recommend the publication of the revised version.